# Clus-UCB: A Near-Optimal Algorithm for Clustered Bandits

**Aakash Gore**  *aakash.gore@iitb.ac.in*
*Department of Electrical Engineering*
*Indian Institute Of Technology Bombay*

**Prasanna Chaporkar**  *chaporkar@ee.iitb.ac.in*
*Department of Electrical Engineering*
*Indian Institute Of Technology Bombay*

**Reviewed on OpenReview:** *https://openreview.net/forum?id=QDMvPO9WJT*

## Abstract

We study a stochastic multi-armed bandit setting where arms are partitioned into known clusters, such that the parameters of arms within a cluster differ by at most a known threshold. While the clustering structure is known a priori, the arm parameters are unknown. We derive an asymptotic lower bound on the regret that improves upon the classical bound of Lai & Robbins (1985). We then propose Clus-UCB, an efficient algorithm that closely matches this lower bound asymptotically by exploiting the clustering structure and introducing a new index to evaluate an arm, which depends on other arms within the cluster. In this way, arms share information among each other. We present simulation results of our algorithm and compare its performance against KL-UCB and other well-known algorithms for bandits with dependent arms. We discuss the robustness of the proposed algorithm under misspecified prior information, address some limitations of this work, and conclude by outlining possible directions for future research.

## 1 Introduction

The multi-armed bandit (MAB) is a fundamental problem in probability theory that encapsulates the classic trade-off between *exploration* and *exploitation*. It is typically abstracted as a scenario in which a gambler is faced with $k$ slot machines (arms), each with an unknown reward distribution, and must decide which arm to pull at each timestep to maximize the cumulative reward. Arms are assumed to belong to the same distribution family, but with different (and unknown) parameters.

A seminal contribution in this area is by Lai & Robbins (1985), who showed that any uniformly good algorithm [1]must incur at least $O(\log N)$ regret, where $N$ is the horizon. Several algorithms such as KL-UCB, UCB, and $\varepsilon$-greedy have been proposed that asymptotically attain this logarithmic regret. This framework models arms that are independent of each other.

Bandit problems where arms are correlated or dependent have also been studied in the literature. These fall into the category of structured bandit problems. Many times, information about the structure results in fewer suboptimal arm pulls and results in lower regret bounds. In this paper, we work with a similar structured bandit problem, specifically one in which arms are clustered together.

### 1.1 Related Work

The classical MAB problem has received significant attention in the past, with one of the most notable contributions being by Lai & Robbins (1985). Using a change-of-measure argument, they derived theoretical

---

[1]A uniformly good algorithm is one which incurs $o(N^a)$ regret for all $a > 0$ on all instances

lower bounds on the regret incurred by comparing an algorithm's performance on two similar instances that differ only in their optimal arms. They also proposed a framework for constructing asymptotically efficient algorithms that achieve logarithmic regret.

A closely related work is that of Graves & Lai (1997), where regret bounds were established for bandits in a controlled Markov chain setting. This work generalizes the procedure of finding a regret lower bound as a linear optimization problem. We use this approach in Section 3 to derive the lower bound for our problem.

For the classical MAB setting with independent arms, several algorithms have been proposed to achieve optimal regret asymptotically. Among the most influential are UCB by Auer et al. (2002), and KL-UCB by Garivier & Cappé (2011). KL-UCB works by selecting the arm with the most optimistic estimate of the mean reward, derived from a KL-divergence-based upper confidence bound. Our proposed algorithm is inspired by this principle and extends it to settings that showcase clustering.

Structured bandits, where dependencies among arms are leveraged to minimize regret, have also been explored. For example, Combes & Proutiere (2014) and Magureanu et al. (2014) studied bandits under uni-modal and Lipschitz structures, respectively, and developed near-optimal algorithms. Mersereau & Tsitsiklis (2009) and Dani et al. (2008) considered linear bandits, where rewards are assumed to be linear functions of unknown parameters. Zhang et al. (2023) studied the MAB problem on a graph, where an agent has to maximize the cumulative reward collected from the nodes of a known graph. Agrawal et al. (1989) studied the case of controlled IID processes with a known finite parameter space, and drew parallels between this and a specialized MAB setting.

Singh et al. (2020) studied a problem setting in which arms are clustered and all arms in a cluster have the same parameter. They assumed that the distributions of rewards for arms in a cluster can be different, depending on the cluster parameter. However, we assume that the parameters of arms in a cluster are within a known threshold, with the same distribution family for all arms in the cluster. Gupta et al. (2021) worked on correlated bandits, where the expected reward of an arm given the reward of another arm is upper-bounded by a known function. This upper bound might be available through domain knowledge or offline estimates.

Pandey et al. (2007) investigated bandits with dependent arms, specifically instances where arms are organized into clusters. They assumed that arm parameters in a cluster are drawn from a known generative model. They formulated a two-level policy assuming that the parameter distribution is tightly centered around its mean. Our problem formulation is a special case of this, where the parameter distribution is uniform over a predefined range, making it spread out rather than tightly centered. It can be seen from the simulation results in Section 5 that the algorithm proposed by them would fail in our setting. This motivates the need for a new algorithm, specific to our case.

Another line of work is dedicated to the online clustering of arms. Ban & He (2021) and Gentile et al. (2014) studied clustering in contextual bandits and provided algorithms that confidently cluster users by the proximity of their parameter vectors. Note that this is distinct from our setting, which assumes known clusters.

To our knowledge, our clustering criterion has not been explored in the literature and is useful if confident cluster width estimates are available through domain knowledge or offline data sampling.

## 1.2 Our Contributions:

- We introduce a framework where arms are organized into *constrained overlapping clusters*, and derive theoretical lower bounds on regret in this structured bandit setting. By constrained, we mean that the arm parameters within a cluster cannot differ by more than a known threshold.

- We propose **Clus-UCB**, an algorithm that efficiently exploits this structure and asymptotically achieves the regret lower bound on most instances.

- We provide both the theoretical analysis of the algorithm's performance in the Appendix, and simulation results in a later section, that demonstrate the practical effectiveness and theoretical optimality of our algorithm.

## 2 Model and Problem Formulation

In this section, we first describe the standard stochastic bandit framework, followed by the specific structure of clustered arms that we address in this work.

### 2.1 Stochastic Bandit Framework

At each round $n = 1, 2, \ldots T$, a learner selects one of $K$ arms and receives a reward sampled from an unknown distribution. Each arm $k$ is associated with an unknown parameter $\theta_k \in \Theta$ and a known density $f(x; \theta_k)$ with respect to a measure $\nu$. We assume:

$$\int |x| f(x; \theta) \, d\nu(x) < \infty, \quad \forall \theta \in \Theta.$$

The expected reward for arm $k$ is given by:

$$\mu_k = \mu(\theta_k) = \int x f(x; \theta) \, d\nu(x).$$

**Policy:** A sequence of arms $\pi = (\pi_n)$, where $\pi_n \in \{1, \ldots, K\}$, where $\pi_n$ is $\mathcal{F}_{n-1}$-measurable (depends only on past actions and rewards).

Let $\mu^* = \max_k \mu_k$ and denote by $T_k^\pi(n)$ the number of times arm $k$ is pulled up to round $n$ under policy $\pi$.

**Regret:** Regret under policy $\pi$ until round $n$ is:

$$R^\pi(n, \nu(\theta)) = \sum_{k:\mu_k < \mu^*} (\mu^* - \mu_k) \mathbb{E}[T_k^\pi(n)].$$

Here, $\theta$ is the parameter vector and $\nu(\theta)$ is the instance.

### 2.1.1 KL Divergence

The Kullback-Leibler(KL) divergence is a measure of difference between two distributions. In bandit problems, this arises naturally to account for the complexity of the problem. If two arm distributions are similar, then it is harder to gauge the optimal arm among these. For densities parameterized by $\theta$ and $\vartheta$:

$$I(\theta, \vartheta) = \int \log\left(\frac{f(x; \theta)}{f(x; \vartheta)}\right) f(x; \theta) \, d\nu(x).$$

The one-sided KL divergence is defined as:

$$I^+(\theta, \vartheta) = \begin{cases} I(\theta, \vartheta) & \text{if } \mu(\theta) < \mu(\vartheta), \\ 0 & \text{otherwise.} \end{cases}$$

**Assumptions:**

- $0 < I(\theta, \vartheta) < \infty$ if $\mu(\vartheta) > \mu(\theta)$,

- $I(\theta, \vartheta)$ is continuous in $\mu(\vartheta)$.

For Bernoulli distributions with parameters $\theta$ and $\vartheta$,

$$I(\theta, \vartheta) = \theta \log\left(\frac{\theta}{\vartheta}\right) + (1 - \theta) \log\left(\frac{1 - \theta}{1 - \vartheta}\right).$$

For the rest of this work, we analyze settings where there is a one to one relation between $\theta_k$ and $\mu_k$, and hence interchangeably use $I(\theta_a, \theta_b)$ and $I(\mu_a, \mu_b)$ when the context is clear.

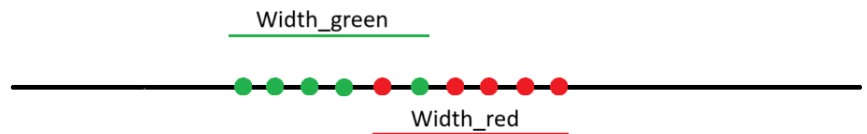

Figure 1: The widths represent cluster spans. While the fifth green point also lies within the red span, it's labeled green. Similarly, the first red point also falls within the green span but is labeled red.

### 2.1.2 KL-UCB Algorithm

For each arm $k$, define the KL-UCB in the $n^{\text{th}}$ round as:

$$\sup\{\vartheta : T_k(n) \cdot I(\hat{\theta}_k(n), \vartheta) \leq \log n + a \log \log n\},$$

where $\hat{\theta}_k(n)$ is the empirical parameter estimate of $k^{\text{th}}$ arm in the $n^{\text{th}}$ round, and $a$ is a constant greater than 3. At each round, select the arm with the highest KL-UCB. Note that each arm must be pulled at least once, for the empirical means to be defined.

### 2.2 Clustered Arm Structure

We now introduce the clustering structure in which arms are grouped into overlapping clusters. Throughout this work, we index the $M$ clusters by $c \in \{1, 2, \ldots, M\}$. Each cluster $c$ contains $K_c$ arms, which we index by $i \in \{1, 2, \ldots, K_c\}$, and we denote the set of arms in cluster $c$ by $\mathcal{K}_c$. For simplicity, we assume there is a unique optimal arm.

For any two arms $i, j$ belonging to the same cluster $c$ with parameters $\theta_c^i$ and $\theta_c^j$ respectively, we require:

$$|\theta_c^i - \theta_c^j| < \beta_c, \quad \text{for all } i, j \in \mathcal{K}_c, \quad c \in \{1, \ldots, M\},$$

where $\beta_c > 0$ is the known cluster width for cluster $c$. We define $\Theta$ as the set of all parameter vectors satisfying this clustering condition.

The assumption of known cluster widths is not purely theoretical but also practical when a rough estimate or non-trivial upper bound on $\beta_c$ is available through domain knowledge or offline sampling. This framework applies naturally to settings where arms share similar attributes. For instance, in online advertising, products from the same category (e.g., electronics) can be assumed to cluster together. Similarly, in clinical trials, drugs with similar chemical compositions form natural clusters. While domain knowledge provides guidance on cluster membership, the widths $\beta_c$ must be estimated from offline data. One possible estimator is:

$$\hat{\beta}_c = \max_{i \in \mathcal{K}_c} u(\theta_c^i) - \min_{i \in \mathcal{K}_c} l(\theta_c^i),$$

where $u(\cdot)$ and $l(\cdot)$ are appropriately chosen upper and lower confidence bound functions, respectively.

Importantly, each arm may satisfy the clustering property for multiple clusters. For example, an arm may fall within the allowed parameter ranges of two different clusters simultaneously. However, each arm is assigned to exactly one cluster, and this assignment is known. This overlapping structure is illustrated in Figure 1.

## 3 Lower Bound for Regret

In this section, we state an asymptotic regret lower bound satisfied by any uniformly good algorithm $\pi$. Let $\nu(\theta)$ be an instance dependent on the parameter vector $\theta$, and let $R^\pi(T, \nu(\theta))$ be the regret incurred by $\pi$ over horizon $T$ on this instance. The algorithm is uniformly good if the regret $R^\pi(T, \nu(\theta)) = o(T^\alpha)$ as $T$ grows large for all $\alpha > 0$ and for all $\theta \in \Theta$.

**Theorem 1:** Let $\pi \in \Pi$ be a uniformly good rule. For any $\theta \in \Theta$, we have:

$$\liminf_{T \to \infty} \frac{R^\pi(T, \nu(\theta))}{\log T} \geq C(\theta),$$

where

$$C(\theta) = \sum_{c=1, c \neq c^*}^{M} \min \left( \sum_{k \in \mathcal{K}_c} \frac{\mu^* - \mu_c^k}{\alpha_c^k L_c}, \frac{\mu^* - \mu_c^{min}}{I^+(\theta_c^{min}, \theta^* - \beta_c)} \right) + \sum_{k \in \mathcal{K}_{c^*} \setminus \{k^*\}} \frac{\mu^* - \mu_{c^*}^k}{I^+(\theta_{c^*}^k, \theta^*)},$$

- $\mu^* = \max_{k,c} \mu(\theta_c^k)$, $\theta^* = \max_{k,c} \theta_c^k$

- $\mu_c^{min} = \min_k \mu(\theta_c^k)$, $\theta_c^{min} = \min_k \theta_c^k$

- $\alpha_c^k = I^+(\theta_c^k, \theta^*) - I^+(\theta_c^k, \theta^* - \beta_c)$

- $b_c^k = I^+(\theta_c^k, \theta^* - \beta_c)$

- $L_c = 1 + \sum_{k=1}^{K_c} \frac{b_c^k}{\alpha_c^k}$

The lower bound of regret is derived using the results of controlled Markov chains from Graves & Lai (1997). The proof outline is presented in Appendix A.2. In general, the main idea used for deriving lower bounds is to consider an alternate instance, which is 'close' to the actual instance under consideration, but has a different optimal mean. The algorithm needs to explore sufficiently to distinguish between the original instance and the alternate instance. For the clustered case under consideration, the alternate instance parameters also belong to the clustered parameter space, unlike the classical case, where the parameters were unconstrained. This fact results in a better lower bound in the structured case. In the proof, we consider an alternate instance as mentioned earlier, and the terms in the lower bound arise naturally as a result of the cluster constraints. The following are some important points:

- The exploration term for an arm, i.e., $\alpha_c^k L_c$, depends on the parameters of other arms in that cluster. This is in contrast to the classical regret bound, where this term only depends on that arm's parameter.

- This regret bound is always lower than that of the classical bandits derived in Lai & Robbins (1985), which is

$$\liminf_{T \to \infty} \frac{R^\pi(T, \nu(\theta))}{\log T} \geq \sum_{c \neq c^*}^{M} \sum_{k \in \mathcal{K}_c} \frac{\mu^* - \mu_c^k}{I(\theta_c^k, \theta^*)} + \sum_{k \in \mathcal{K}_{c^*} \setminus \{k^*\}} \frac{\mu^* - \mu_{c^*}^k}{I(\theta_{c^*}^k, \theta^*)}$$

- For arms belonging to $c^*$, the regret term is the same as that of classical bandits. This is because inside $c^*$, the cluster structure makes no difference. On the other hand, for suboptimal clusters, we exploit the structure and make improvements in the bound.

- It is seen that the regret contains a $\min(a, b)$ term. The second argument in this corresponds to the regret incurred by only pulling the worst arm in a cluster. All other arms in the cluster must be pulled sub-logarithmic times in expectation. Intuitively, the second term corresponds to instances where it is relatively easier to distinguish the minimum arm in a cluster from the best arm in the instance, while the first term corresponds to instances where the agent must pull all arms in the cluster to be certain of its suboptimality. Hence, it is likely that if we have a loosely constrained cluster, the first term would be the minimum, while for tight clusters, the second term would be the minimum. However, in most of our presented here or otherwise, we found that the first term appears in the lower bound.

- Note that for the trivial case of $\beta_c = 1$ for Bernoulli bandits, we essentially have no clustering information. Thus, the lower bound term for that cluster becomes the same as in the classical case. Here, we slightly abuse notation to convey that

$$\frac{\mu^* - \mu_c^{min}}{I^+(\theta_c^{min}, \theta^* - \beta_c)} = \frac{\mu^* - \mu_c^{min}}{0} = \infty,$$

and

$$\sum_{k \in \mathcal{K}_c} \frac{\mu^* - \mu_c^k}{\alpha_c^k L_c} = \sum_{k \in \mathcal{K}_c} \frac{\mu^* - \mu_c^k}{I(\theta_c^k, \theta^*) + \sum_{k' \in c, k' \neq k} I^+(\theta_c^{k'}, \theta^* - \beta_c)} = \sum_{k \in \mathcal{K}_c} \frac{\mu^* - \mu_c^k}{I(\theta_c^k, \theta^*)}.$$

- For the trivial case of $\beta_c = 0$, we have $K_c$ arms with the same parameter. Hence, $\theta_c^{min} = \theta_c^k = \theta$(let). So, $\mu_c^{min} = \mu_c^k = \mu(\theta) = \mu$(let).This essentially means that we have only one arm in the cluster.

$$\frac{\mu^* - \mu_c^{min}}{I^+(\theta_c^{min}, \theta^* - \beta_c)} = \frac{\mu^* - \mu}{I^+(\theta, \theta^*)},$$

and

$$\sum_{k \in \mathcal{K}_c} \frac{\mu^* - \mu_c^k}{\alpha_c^k L_c} = \sum_{k \in \mathcal{K}_c} \frac{\mu^* - \mu}{I(\theta, \theta^*) + \sum_{k' \in c, k' \neq k} I(\theta, \theta^*)} = \frac{\mu^* - \mu}{I(\theta, \theta^*)}.$$

## 4 Clus-UCB Algorithm

We now present an algorithm whose regret closely matches the lower bound derived in the previous section for clustered overlapping bandits.

---

**Algorithm 1** Clus-UCB Algorithm

---

**Require:** Horizon $T$, number of clusters $M$, arm sets $\{\mathcal{K}_c\}_{c=1}^M$ with $|\mathcal{K}_c| = K_c$, total number of arms $K$, cluster gap parameters $\{\beta_c\}_{c=1}^M$, constant $a \geq 5$

1: Pull each arm once; initialize $t_c^k(K+1) = 1$ and estimates $\hat{\theta}_c^k(K+1)$
2: **for** $n = K + 1$ to $T$ **do**
3:     **for** each cluster $c = 1, \ldots, M$ and each arm $k \in \mathcal{K}_c$ **do**
4:         Update empirical estimate $\hat{\theta}_c^k(n)$
5:     **end for**
6:     **for** each cluster $c = 1, \ldots, M$ and each arm $k \in \mathcal{K}_c$ **do**
7:         Compute Clus-UCB index:

$$v_c^k(n) = \sup \left\{ q : t_c^k(n) I^+\left(\hat{\theta}_c^k(n), q\right) + \sum_{k' \in \mathcal{K}_c \setminus \{k\}} t_c^{k'}(n) I^+\left(\hat{\theta}_c^{k'}(n), q - \beta_c\right) \leq \log n + a \log \log n \right\}$$

8:     **end for**
9:     **if** $\exists (c, k)$ such that $t_c^k(n) \leq \log \log n$ **then**
10:         Pull arm $k$ in cluster $c$
11:     **else**
12:         Select $(c', k') = \arg\max_{c,k} v_c^k(n)$
13:         Pull arm $k'$ in cluster $c'$
14:     **end if**
15:     Observe reward and update counts
16: **end for**

---

**Theorem 2:** Assuming that the bandit arms have bounded rewards or belong to the canonical exponential family $p_\theta(x) = \exp(x\theta - b(\theta) + c(x))$, with parameters clustered according to Section 2.2, Clus-UCB's asymptotic regret is upper-bounded as

$$\liminf_{T \to \infty} \frac{R^\pi(T, \nu(\theta))}{\log T} \leq C(\theta),$$

where

$$C(\theta) = \sum_{c=1, c \neq c^*}^M \sum_{k \in \mathcal{K}_c} \frac{\mu^* - \mu_c^k}{\alpha_c^k L_c} + \sum_{k \in \mathcal{K}_c \setminus \{k^*\}} \frac{\mu^* - \mu_{c^*}^k}{I^+(\theta_{c^*}^k, \theta^*)}.$$

The proof of Theorem 2 for Bernoulli bandits is present in Appendix A.3. This can be extended to any family with bounded rewards according to Lemma 9 and Theorem 10 in Garivier & Cappé (2011). Theorem 2 in Magureanu et al. (2014) implies that this holds for the canonical exponential family.

The following are some key points about this algorithm:

- **Forced Exploration of Under-Explored Arms:** The algorithm design ensures that an arm with less than $\log\log n$ pulls at time $n$ gets pulled eventually. This ensures that no arm faces starvation and all arms get pulled infinitely often.

- **Rare Suboptimal Pulls Due to Confidence Underestimation:** The event in which an arm of a suboptimal cluster is pulled because the Clus-UCB of the optimal arm falls below its mean occurs only $O(\log\log T)$ times.

- **Pull Ratio Among Arms in a Suboptimal Cluster:** Within a suboptimal cluster, arms are pulled in inverse proportion to their exploration coefficients . That is, for two arms with exploration parameters $\alpha_{k_1}^c$ and $\alpha_{k_2}^c$, the expected number of times they are pulled over a long time satisfies:

$$E[t_c^{k_1}] : E[t_c^{k_2}] \approx \alpha_c^{k_2} : \alpha_c^{k_1}$$

- **Expected Pulls of Arms in Suboptimal Clusters:** An arm $k$ belonging to a suboptimal cluster $c$ is pulled approximately

$$\frac{\log T}{\alpha_c^k \cdot L_c} + O(\log\log T).$$

  times in expectation, over a long time, where $\alpha_c^k$ and $L_c$ are as defined earlier.

- **Near-Optimality:** The upper bound presented above, matches the regret lower bound derived earlier on most instances, but not all. This makes the algorithm near-optimal.

The motivation to use the Clus-UCB index as done in the algorithm is through the lower bound derived and the analysis done by Garivier & Cappé (2011). A forced exploration term is also added to ensure all arms get pulled infinitely often.

Intuitively, the optimality of an arm is evaluated not only by its individual performance but also through the collective behavior of its cluster. When other members of the same cluster provide sufficient evidence indicating that the cluster as a whole is suboptimal, the index of an arm is accordingly suppressed even if its own empirical mean remains high, thereby reducing unnecessary exploration.

The regret lower bound formally captures this interdependence, justifying the proposed index as a theoretically motivated choice. Specifically, the index quantifies the exploration required to distinguish the current instance from one in which arm $k$ in cluster $c$ is optimal. The term corresponding to arm $k$ coincides with the standard KL-UCB term, while the additional $I^+(\hat{\theta}_c^{k'}, q - \beta_c)$ components represent the information contributions from other arms within the same cluster, under the hypothesis that arm $k$ were optimal. The offset term $q - \beta_c$ incorporates the effective range of these dependencies.

## 5    Simulation Results and Discussion:

We ran simulations comparing KL-UCB, Clus-UCB and a KL-UCB-based Two-level-Policy(TLP) on different bandit instances, and figures 2-8 show the results. For simplicity, the simulations omit the forced exploration component (lines 9-10 of the algorithm). Given our time horizon of $10^6$ steps, this omission has negligible impact on performance. All experiments were performed on a computer with 16 gigabytes of RAM. No GPU was used. The plots show the results of 200 simulations, with the means and the 95% confidence intervals presented. To speed up the simulations, we used a multiprocessing framework with 16 CPU cores. Furthermore, we updated the UCBs every 50 timesteps to reduce computation time. The UCBs were calculated using binary search, and are accurate up to 4 decimal places. Here, $\beta$ is the cluster width vector.

We consider two variants of the Two-Level Policy (TLP) suggested by Pandey et al. (2007): MEAN and MAX. TLP treats each cluster as a "super arm" and uses a base policy (KL-UCB) to choose which cluster to play. Once a cluster is selected, the base policy is applied to its arms. Cluster selection requires a cluster reward estimate:

- In MEAN, this is the total successes of all arms in the cluster divided by the total cluster pulls.

$$\hat{r}_c^{MEAN}(n) = \frac{\sum_{k \in \mathcal{K}_c} t_c^k(n) \hat{\mu}_c^k(n)}{\sum_{k \in \mathcal{K}_c} t_c^k(n)}$$

- In MAX, it is the maximum empirical mean among the cluster's arms

$$\hat{r}_c^{MAX}(n) = \max \hat{\mu}_c^k(n)$$

In our experiments, Clus-UCB consistently outperforms KL-UCB. However, on certain instances, TLP can outperform Clus-UCB. That said, TLP is not asymptotically optimal, as it lacks knowledge of cluster widths. Moreover, since TLP assumes arm parameters are tightly clustered, it is straightforward to construct hard instances where its performance degrades sharply (see Figure 5). This is because on average, the suboptimal cluster will have a higher cluster estimate as compared to the optimal cluster, which results in the suboptimal cluster being pulled more often. We also ran a simulation with 20 clusters and 55 arms, incorporating a wide variety of cluster types together. The result can be found in the Appendix A.1.

## 5.1 TLP vs Clus-UCB

The simulation results demonstrate that TLP either significantly outperforms Clus-UCB or performs similarly when clusters are non-overlapping, while Clus-UCB significantly outperforms TLP for heavily overlapping clusters. This performance difference can be explained theoretically: for instances with non-overlapping clusters, the regret lower bound is independent of cluster widths and is lower than the bound presented in Section 3. In such cases, Clus-UCB's design makes it overly conservative during exploitation, leading to unnecessary exploration beyond what is required. TLP, however, does not suffer from this issue, as the non-overlapping structure allows the cluster reward estimates to accurately identify the optimal cluster most times.

Therefore, we recommend using TLP when clusters are known to be non-overlapping. However, when such well-separation assumptions cannot be guaranteed, Clus-UCB is the preferred choice, as it leverages cluster width information to achieve more efficient exploration in these challenging scenarios.

## 6 Misspecification of Cluster Widths

An important point to consider is the misspecification of cluster widths. Cases where the exact widths are not known, but an estimate is available, might be more practical. If the widths are overestimated, the proposed algorithm continues to outperform KL-UCB. The case of underestimated widths, however, is more nuanced. In the proof of Clus-UCB's optimality (Appendix), we divide the total number of pulls of an arm in a suboptimal cluster into two cases:

- when the Clus-UCB index of the optimal arm is less than its mean, and

- when the index is greater than or equal to its mean.

We bound these two terms separately. It is noteworthy that the cluster constraint is used only in bounding the first term, which leads to an $O(\log \log T)$ bound. Moreover, this bound depends solely on the width of the optimal cluster. In fact, throughout the proof, there is no requirement that the other (suboptimal) clusters satisfy their respective constraints.

At first glance, this may appear surprising. However, the problem formulation we present is actually a special case of a more general setting of the allowed instances: every cluster has an associated width, but the cluster

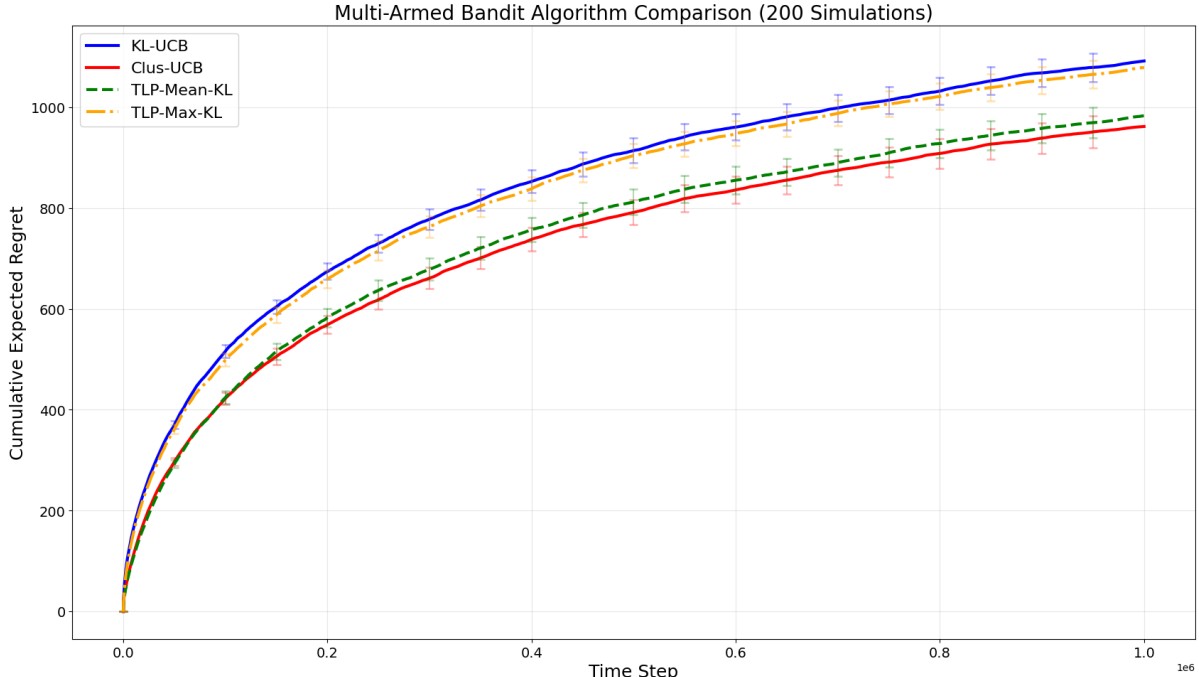

Figure 2: Comparison of Clus-UCB with other algorithms on the instance [0.40,0.41,0.42], [0.60,0.61,0.62] with $\beta = [0.02, 0.02]$ and a horizon of $10^6$ time steps. These represent well separated clusters. The first term in min appears in the regret lower bound for the suboptimal cluster in this instance.

constraint is required to hold only for the optimal cluster in the fixed instance, while suboptimal clusters may violate it. Our formulation imposes the stricter condition that all clusters satisfy their constraints, which is a subset of the general case. This is distinct from the scenario where, in a given instance, a fixed cluster (which happens to be optimal) satisfies the constraint but an originally suboptimal cluster which becomes optimal in an alternate instance need not satisfy this constraint. In the general setting, while constructing an alternate instance, the originally suboptimal cluster may become optimal and must then satisfy its width constraint. In our setting, the regret lower bound derived earlier continues to apply to this more general case as well.

From this perspective, underestimating the width of a suboptimal cluster does not harm performance—in fact, it improves the performance due to the larger denominator ($\hat{\beta}_c < \beta_c$) in the regret bound. This is shown in Figures 4 and 6, where the cluster width estimate for the suboptimal cluster is reduced from correctly estimated to underestimated. However, underestimating the width of the optimal cluster can lead to substantial regret. This is because the $O(\log \log T)$ bound may not hold now. This is shown in Figure 7.

If all cluster widths are specified correctly or are overestimated, Clus-UCB retains the property of being uniformly good. Hence, for the algorithm to work well, the sufficient condition is that the optimal cluster must have an overestimated width.

# 7 Limitations and Future Work

The algorithm provided is asymptotically optimal for most instances, but not all. However, we believe that a more carefully chosen optimistic index, might perform optimally on all instances, albeit with increased complexity of analysis. It is also possible to develop a randomized Bayesian algorithm, similar to Thompson sampling. The beliefs would still be Beta distributed, but only supporting parameter values that satisfy the clustering constraint. We leave the proof of optimality of this algorithm for future work. Finally, though overestimation of the optimal cluster's width does not harm much, underestimating it may be significantly

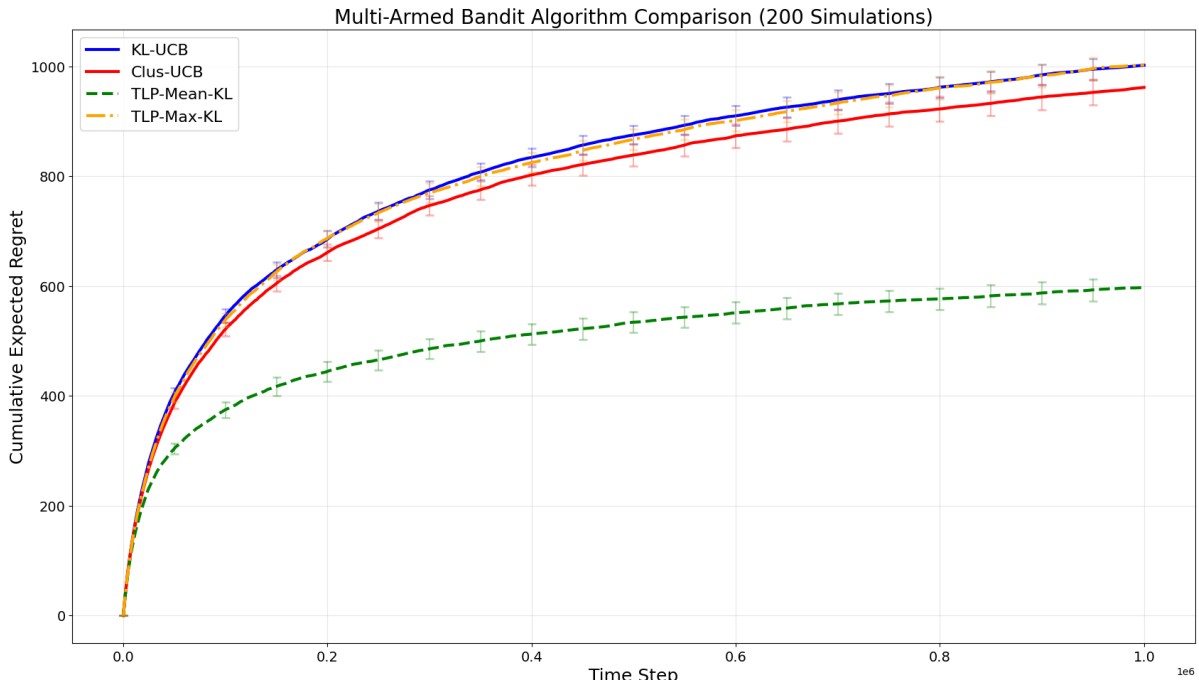

Figure 3: Comparison of Clus-UCB with other algorithms on the instance [0.80,0.82,0.84], [0.81,0.83,0.85] with $\beta = [0.04, 0.04]$ and a horizon of $10^6$ time steps. These represent overlapping clusters. The first term in min appears in the regret lower bound for the suboptimal cluster in this instance.

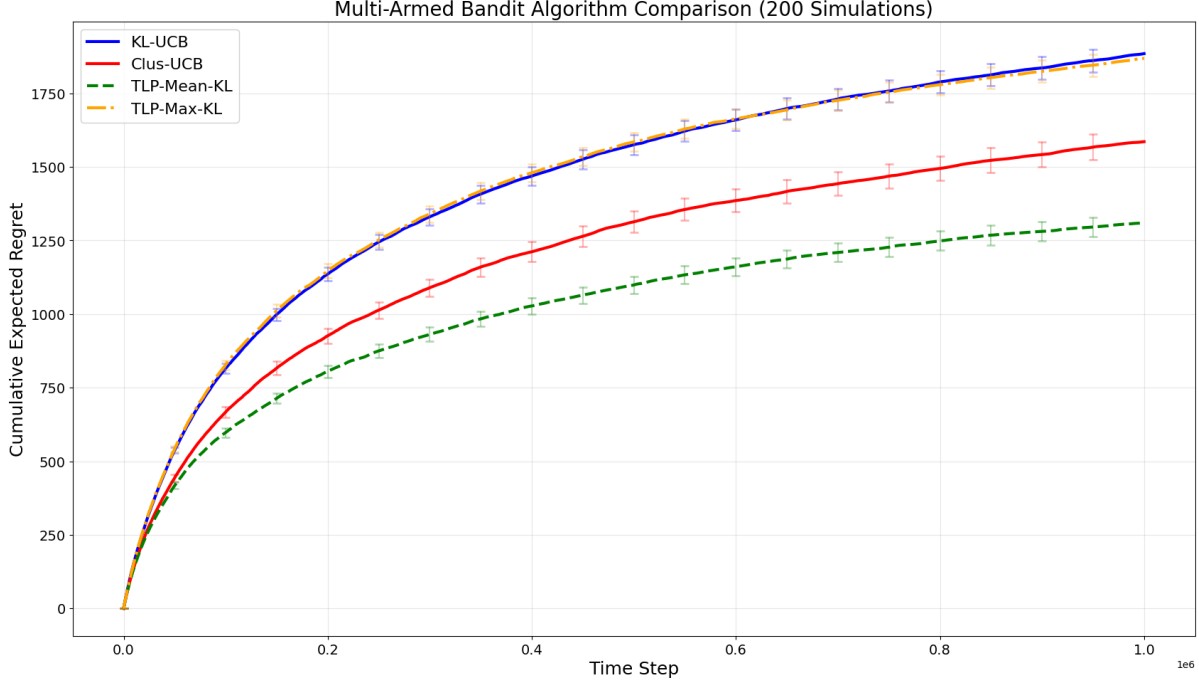

Figure 4: Comparison of Clus-UCB with other algorithms on the instance [0.41,0.42,0.43], [0.43,0.44,0.45] with $\beta = [0.02, 0.02]$ and a horizon of $10^6$ time steps. These represent close but separated clusters. The first term in min appears in the regret lower bound for the suboptimal cluster in this instance.

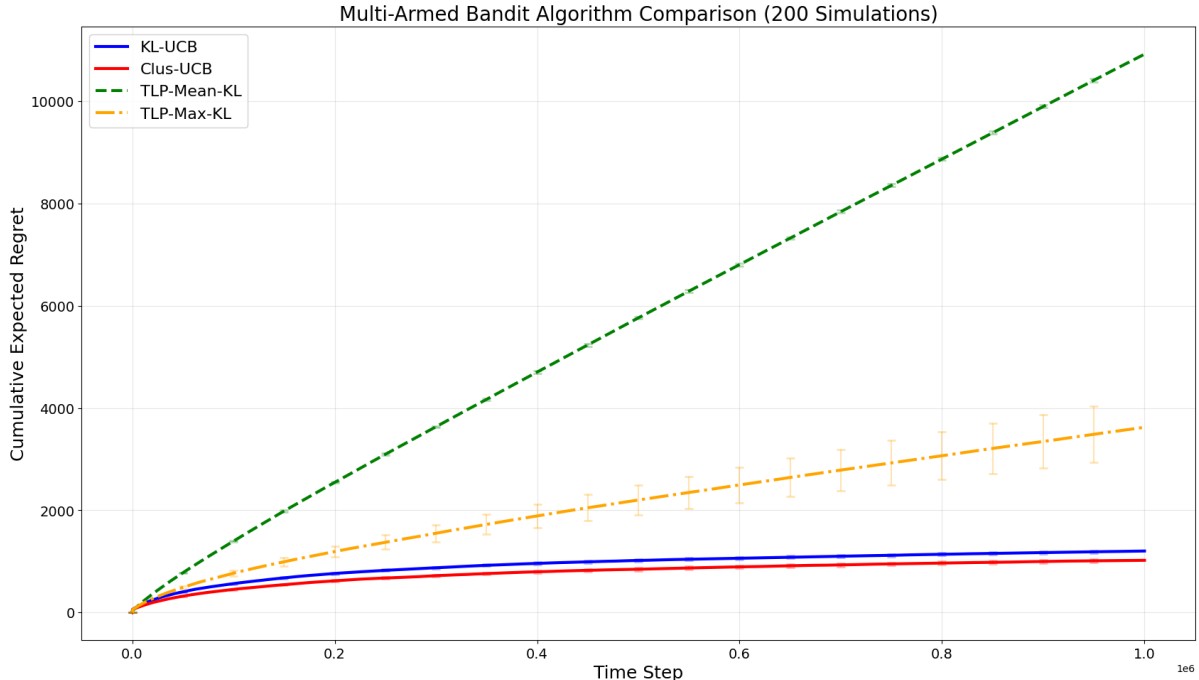

Figure 5: Comparison of Clus-UCB awith other algorithms on the instance [0.68,0.69,0.67], [0.1,0.2,0.7] with $\beta = [0.02, 0.8]$ and a horizon of $10^6$ time steps. This represents an instance where the TLP-Mean policy performs poorly. The first term in min appears in the regret lower bound for the suboptimal cluster in this instance.

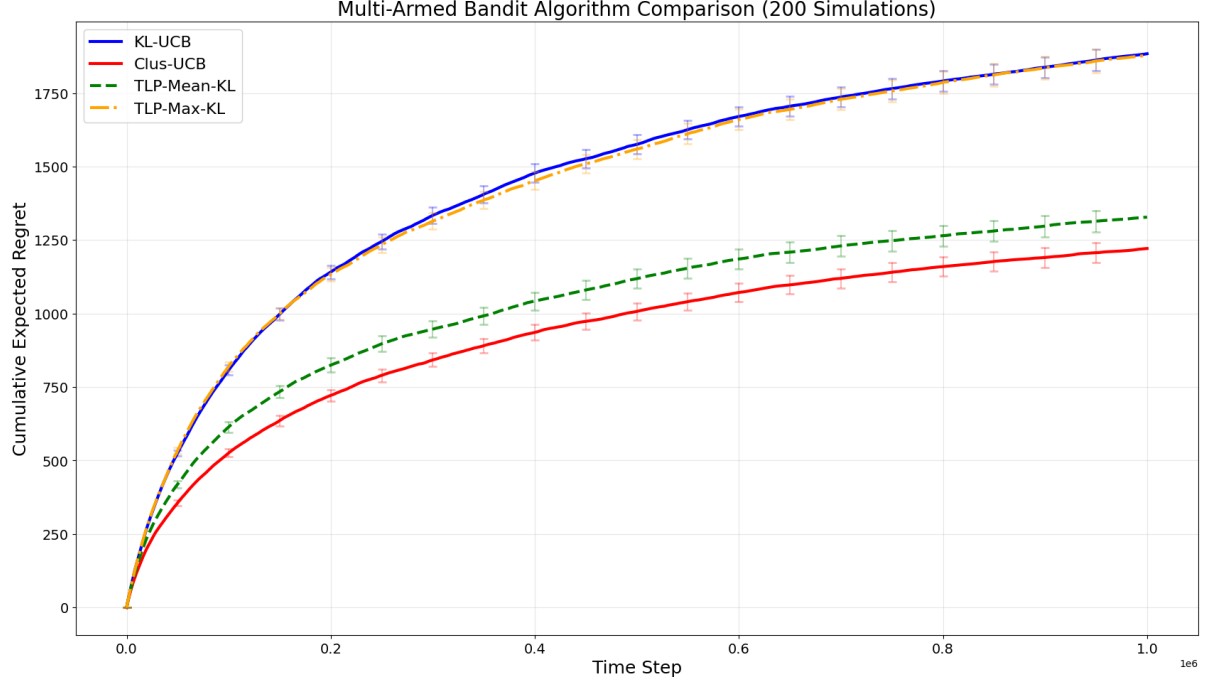

Figure 6: Comparison of Clus-UCB with other algorithms on the instance [0.41,0.42,0.43], [0.43,0.44,0.45] with $\beta = [0.00, 0.02]$ and a horizon of $10^6$ time steps. These represent close but separated clusters. The second term in min appears in the regret lower bound for the suboptimal cluster in this instance.

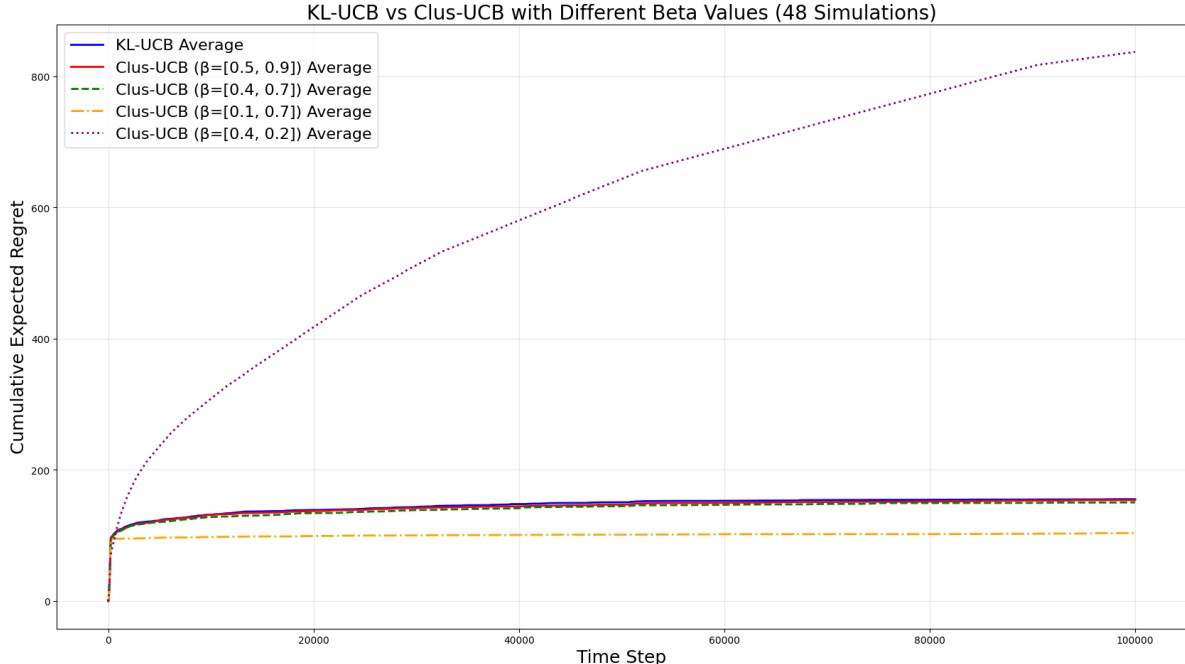

Figure 7: Comparison of Clus-UCB and KL-UCB on the instance $[0.3,0.7],[0.1,0.2,0.8]$ with $\beta = [0.5, 0.9]$, $\beta = [0.4, 0.7]$, $\beta = [0.1, 0.7]$, and $\beta = [0.4, 0.2]$ and a horizon of $10^6$ time steps.

harmful. Thus, a separate framework dealing with width estimates as unreliable side-information seems worth exploring.

## 8 Conclusion

In this work, we analyzed a clustered bandit setting where prior information on an upper bound of cluster widths is available. We established an improved regret lower bound compared to the classical result of Lai & Robbins (1985) for unstructured bandits, and proposed the Clus-UCB algorithm to exploit this structure. Our analysis showed its near-optimality, and simulations confirmed its advantages over structure-unaware algorithms as well as its competitiveness with the two-level policy of Pandey et al. (2007). We further examined cases where cluster widths are misspecified and identified a necessary condition for Clus-UCB to remain robust under such settings. Despite these contributions, certain limitations remain. In particular, our algorithm is near-optimal, however, a more carefully chosen index might lead to an optimal algorithm.

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

## A  Appendix

### A.1  Extra Simulation Results:

The instance under consideration was [0.1,0.7,0.9], [0.1,0.15], [0.12,0.14,0.13], [0.23,0.30], [0.28,0.34,0.32], [0.16,0.17], [0.65,0.61,0.63,0.67], [0.1,0.2,0.95], [0.91,0.92], [0.40,0.41,0.42], [0.81,0.86,0.89], [0.78,0.85,0.88], [0.1,0.5,0.9], [0.92,0.93,0.94], [0.01,0.02,0.03], [0.05,0.1], [0.2,0.36], [0.2,0.21,0.78], [0.3,0.29,0.53,0.54], [0.57,0.6] with cluster widths [ 0.8, 0.05, 0.02, 0.07, 0.06, 0.01, 0.05, 0.85, 0.01, 0.02, 0.08, 0.10, 0.8, 0.02, 0.02, 0.05, 0.16, 0.58, 0.25, 0.03 ]

### A.2  Proof of Theorem 1

We follow the analytical framework developed by Graves & Lai (1997). Let $\Theta$ denote the set of all problem instances consistent with the given cluster structure. For each arm $j$, define $\Theta_j$ as the set of instances in which arm $j$ is optimal. Given an instance $\theta \in \Theta$, let $J(\theta)$ be the set of optimal arms under $\theta$.

We define the set of *bad instances* as:

$$B(\theta) = \left\{ \lambda \in \Theta : \mu_\theta^j = \mu_\lambda^j \ \forall j \in J(\theta), \text{ and } \lambda \notin \bigcup_{j \in J(\theta)} \Theta_j \right\}.$$

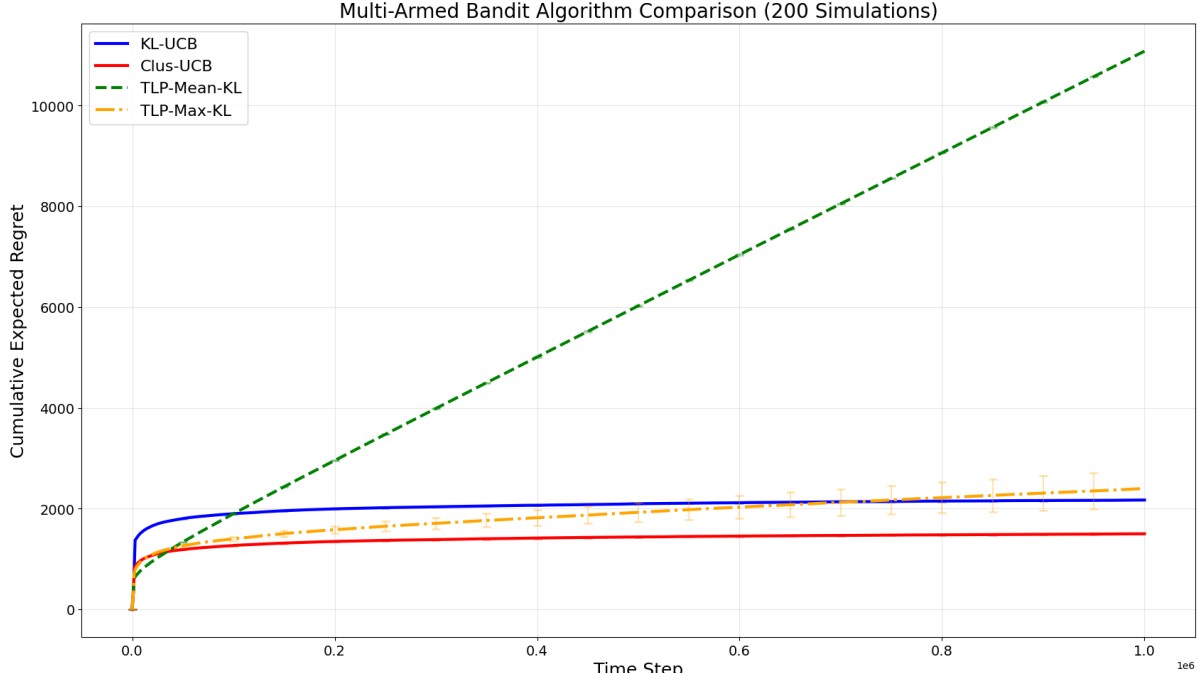

Figure 8: Comparison of Clus-UCB with other algorithms for 20 clusters with 55 arms.

Here $\mu_\theta^j$ is the mean of arm $j$ in the instance $\nu(\theta)$. Let $C(\theta)$ be the value of the following optimization problem:

$$C(\theta) = \inf\left\{ \sum_{j \notin J(\theta)} C_j(\mu_\theta^* - \mu_\theta^j) : C_j \geq 0,\ \inf_{\lambda \in B(\theta)} \sum_{j \notin J(\theta)} C_j I(\mu_\theta^j, \mu_\lambda^j) \geq 1 \right\},$$

where $I(\cdot, \cdot)$ is the KL divergence.

According to Theorem 1 of Graves & Lai (1997), this quantity characterizes the asymptotic lower bound on regret for any uniformly good algorithm $\pi$:

$$\liminf_{n \to \infty} \frac{R^\pi(n, \nu(\theta))}{\log n} = C(\theta).$$

Computing $C(\theta)$ reduces to solving a linear program. Suppose that under a bad instance $\lambda$, some arm $i$ from a suboptimal cluster $c_0$ becomes optimal. The value of

$$\sum_{j \notin J(\theta)} C_j I(\mu_\theta^j, \mu_\lambda^j).$$

is minimized when, for all clusters except $c_0$, the arm means under $\lambda$ match those of the suboptimal arms in $\theta$. For cluster $c_0$, the $i$-th arm has a mean greater than $\mu_\theta^*$, while other arms in $c_0$ have means

$$\mu_\lambda^j = \max(\mu_\theta^j, \mu_\lambda^* - \beta_c).$$

The minimum is achieved when $\mu_\lambda^* = \mu_\theta^*$.

For a given cluster $c_0$ with $K_{c_0}$ arms indexed by $k = 1, \ldots, K_{c_0}$, the system of inequalities becomes:

$$C_i I(\mu_i, \mu^*) + \sum_{k \in \mathcal{K}_{c_0} \setminus \{i\}} C_k I^+(\mu_k, \mu^* - \beta_c) \geq 1,$$

where $\mu_i = \mu_\theta^i$, and $\mu^* = \mu_\theta^*$.

Let $\mathbf{B}$ be a $K_{c_0} \times K_{c_0}$ matrix, where each column $i$ has elements $b_c^i$. Let $\boldsymbol{\alpha}$ be a diagonal matrix of $\alpha_{c_0}^k$ terms, and $\mathbf{c}$ a column vector of the $C_i$ variables. Define the reward gap vector $\mathbf{a}$ with entries $a_i = \mu^* - \mu_i$.

The linear program becomes:

$$\min_{\mathbf{c} \geq 0} \quad \mathbf{a}^\top \mathbf{c}$$
$$\text{subject to} \quad (\mathbf{B} + \boldsymbol{\alpha})\mathbf{c} \geq \mathbf{1}.$$

This optimization can be solved using standard techniques, and we get the desired lower bound.

This process is repeated across all clusters to compute the global infimum $C(\boldsymbol{\theta})$.

## A.3 Proof of Theorem 2

This work follows the outline of the proof of Theorem 2 in Garivier & Cappé (2011). We now state theorem from Magureanu et al. (2014).

**Theorem 3:**

For all $\delta > K + 1$, $n \in \mathbb{N}$, we have:

$$P\left(\sum_{k=1}^{K} t_c^k(n)d^+\left(\hat{\mu}_c^k, \mu_c^k\right) \geq \delta\right) \leq e^{-\delta}\left((\frac{\lceil \delta \log n \rceil \delta}{K})^K e^{K+1}\right).$$

If $\delta_n = \log n + a \log \log n$, with $a \geq 5$, then

$$\mathbb{E}\left[\sum_{n=1}^{T} \mathbb{I}\left\{\sum_{k=1}^{K} t_c^k(n)d^+\left(\hat{\mu}_c^k, \mu_c^k\right) \geq \delta_n\right\}\right] = O(\log \log T)$$

Let $i$ be the best arm in cluster $c$. Let $A_n$ be the arm-cluster pair pulled by the algorithm at time step $n$. Now, we proceed by bounding the number of pulls as:

$$t_c^i(T) = \sum_{n=1}^{T} \mathbb{I}\{A_n = (i,c)\}$$
$$= \sum_{n=1}^{T} \mathbb{I}\left\{A_n = (i,c), v^*(n) \geq \mu^*\right\} + \mathbb{I}\left\{A_n = (i,c), v^*(n) < \mu^*\right\},$$

where $v^*(n)$ is the Clus-UCB of the optimal arm at timestep $n$. Now,

$$\sum_{n=1}^{T} \mathbb{I}\left\{A_n = (i,c), v^*(n) < \mu^*\right\} \leq \sum_{n=1}^{T} \mathbb{I}\left\{v^*(n) < \mu^*\right\}$$

Notice that

$$t_{c^*}^{k^*}(n)d\left(\hat{\mu}_{c^*}^{k^*}(n), \mu^*\right) + \sum_{k \in \mathcal{K}_c \setminus \{k^*\}} t_{c^*}^k(n)d^+\left(\hat{\mu}_{c^*}^k(n), \mu_{c^*}^k\right) \geq t_{c^*}^{k^*}(n)d^+\left(\hat{\mu}_{c^*}^{k^*}(n), \mu^*\right) + \sum_{k \in \mathcal{K}_c \setminus \{k^*\}} t_{c^*}^k(n)d^+\left(\hat{\mu}_{c^*}^k(n), \mu^* - \beta_{c^*}\right)$$

as $u_{c^*}^k \geq u^* - \beta_{c^*} \quad \forall k \in c^*$, and as KL-divergence is increasing in the second term.

Define

$$B_n = t_{c^*}^{k^*}(n)d\left(\hat{\mu}_{c^*}^{k^*}(n), \mu^*\right) + \sum_{k \in \mathcal{K}_c \setminus \{k^*\}} t_{c^*}^k(n)d^+\left(\hat{\mu}_{c^*}^k(n), \mu_{c^*}^k\right)$$

$$C_n = t_{c^*}^{k^*}(n)d^+\left(\hat{\mu}_{c^*}^{k^*}(n), \mu^*\right) + \sum_{k \in \mathcal{K}_c \setminus \{k^*\}} t_{c^*}^k(n)d^+\left(\hat{\mu}_{c^*}^k(n), \mu^* - \beta_{c^*}\right)$$

Hence, $B_n \geq C_n$

Now $P(v^*(n) < \mu*) \leq P(C_n > \log n + a \log \log n) \leq P(B_n > \log n + a \log \log n)$
Hence, using Theorem 3, we have

$$\mathbb{E}[\sum_{n=1}^{T} \mathbb{I}\{v^*(n) < \mu^*\}] \leq \sum_{n=1}^{T} P(C_n > \log n + a \log \log n) \leq \sum_{n=1}^{T} P(B_n > \log n + a \log \log n) = O(\log \log T)$$

**Other term:**
$$\sum_{n=1}^{T} \mathbb{I}\{A_n = (i,c), v^*(n) \geq u^*\}$$

$$= \sum_{n=1}^{T} \mathbb{I}\{A_n = (i,c), v^*(n) \geq u^*, t_c^i(n) \leq \log \log n\} + \sum_{n=1}^{T} \mathbb{I}\{A_n = (i,c), v^*(n) \geq u^*, t_c^i(n) > \log \log n\}$$

$$= O(\log \log T) + \sum_{n=1}^{T} \mathbb{I}\{A_n = (i,c), v^*(n) \geq u^*, t_c^i(n) > \log \log n, \}$$

Note that
$$\{A_n = (i,c), v^*(n) \geq u^*, t_c^i(n) > \log \log n\} \Rightarrow v_c^i(n) \geq v^*(n) \geq u^*$$

Thus, the inequality continues as

$$\leq \sum_{n=1}^{T} X_n . \mathbb{I}\left\{ t_c^i(n) d\left(\hat{\mu}_c^i(n), \mu^*\right) + \sum_{k \in \mathcal{K}_c \setminus \{i\}} t_c^k(n) d^+\left(\hat{\mu}_c^k(n), \mu^* - \beta_c\right) \leq \log n + a \log \log n \right\},$$

where
$$X_n = \mathbb{I}\left\{A_n = (i,c), v_c^i(n) \geq u^*, t_c^i(n) > \log \log n\right\}$$

$$\leq \sum_{n=1}^{T} X_n . \mathbb{I}\left\{ t_c^i(n)[d\left(\hat{\mu}_c^i(n), \mu^*\right) + \sum_{k \in \mathcal{K}_c \setminus \{i\}} \frac{t_c^k(n)}{t_c^i(n)} d^+\left(\hat{\mu}_c^k(n), \mu^* - \beta_c\right)] \leq \log n + a \log \log n \right\}$$

We now make 2 key observations about the behavior of the algorithm:

1. **Regret upper bound:** The regret of the algorithm can be upper bounded by that of the KL-UCB algorithm. This follows as:

$$\sum_{n=1}^{T} \mathbb{I}\left\{ t_c^i(n)[d\left(\hat{\mu}_c^i(n), \mu^*\right) + \sum_{k \in \mathcal{K}_c \setminus \{i\}} \frac{t_c^k(n)}{t_c^i(n)} d^+\left(\hat{\mu}_c^k(n), \mu^* - \beta_c\right)] \leq \log n + a \log \log n \right\}$$

$$\leq \sum_{n=1}^{T} \mathbb{I}\left\{t_c^i(n) d\left(\hat{\mu}_c^i(n), \mu^*\right) \leq \log n + a \log \log n\right\}.$$

   The right hand term is what we get while analyzing KL-UCB. Thus, the regret of Clus-UCB is upper bounded by the regret of KL-UCB. Hence, all suboptimal arms are pulled $O(\log T)$ times in expectation.

2. **Convergence of Clus-UCB values:** The Clus-UCB values of all suboptimal arms must converge to $\mu^*$, the mean of the optimal arm. We prove this by contradiction:

   (a) Suppose the Clus-UCB of a suboptimal arm $i$ converges to some value $u < \mu^*$. Then, eventually, the algorithm will stop selecting this arm. As a result, the number of times it is pulled will be sub-logarithmic, contradicting the earlier claim that every suboptimal arm is pulled $O(\log T)$ times.

(b) Suppose instead that the Clus-UCB of a suboptimal arm converges to $u > \mu^*$. Since the Clus-UCB of the optimal arm converges to $\mu^*$, the suboptimal arm will eventually have a strictly higher UCB. This would lead the algorithm to pull it linearly often, resulting in linear regret, which contradicts the $O(\log T)$ upper bound.

Now, notice that

$$t_c^i(n)d\left(\hat{\mu}_c^i(n), v_c^i(n)\right) + \sum_{k \in \mathcal{K}_c \backslash \{i\}} t_c^k(n)d^+\left(\hat{\mu}_c^k(n), v_c^i(n) - \beta_c\right) = \log n + a \log \log n,$$

$$\Rightarrow t_c^i(n)(d\left(\hat{\mu}_c^i(n), v_c^i(n)\right) - d^+\left(\hat{\mu}_c^i(n), v_c^i(n) - \beta_c\right)) + \sum_{k \in \mathcal{K}_c} t_c^k(n)d^+\left(\hat{\mu}_c^k(n), v_c^i(n) - \beta_c\right) = \log n + a \log \log n.$$

Let $f_i(n) = t_c^i(n)(d\left(\hat{\mu}_c^i(n), v_c^i(n)\right) - d^+\left(\hat{\mu}_c^i(n), v_c^i(n) - \beta_c\right))$ and $g_i(n) = \sum_{k \in \mathcal{K}_c} t_c^k(n)d^+\left(\hat{\mu}_c^k(n), v_c^i(n) - \beta_c\right)$

Thus, $f_k(n) + g_k(n) = \log n + a \log \log n$ for all arms $k$ in cluster $c$. Also, since we are interested in the time instances when arm $i$ has the maximum cluster index, $g_i(n) \geq g_k(n) \forall k \in 1, 2...K_c$ as $v_c^i(n) \geq v_c^k(n)$
This implies, $f_i(n) \leq f_k(n) \forall k \in 1, 2...K_c$. Thus,

$$\frac{t_c^k(n)}{t_c^i(n)} \geq \frac{d(\hat{\mu}_c^i(n), v_c^i(n)) - d^+(\hat{\mu}_c^i(n), v_c^i(n) - \beta_c)}{d(\hat{\mu}_c^k(n), v_c^k(n)) - d^+(\hat{\mu}_c^k(n), v_c^k(n) - \beta_c)}$$

Due to the design of the algorithm(forced exploration of arms with less than $\log \log n$ pulls), all arms are pulled infinitely often asymptotically, i.e, no arm faces starvation. Hence, by the Strong Law of Large Numbers, the empirical mean of each arm converges almost surely to its true mean. This implies that for any $\epsilon > 0$, each arm's empirical mean differs from its true mean by more than $\epsilon$ only finitely many times along its pull sequence. Consequently, almost surely, there exists a random time after which each arm's empirical mean remains $\epsilon$-close to its true mean. For a finite cluster of arms, by taking the maximum of these (finite) random times, we can conclude that almost surely, there exists a finite time $N_\epsilon$ after which all empirical means in the cluster are simultaneously $\epsilon$-close to their true means. Also, $v_c^i(n) \geq \mu*$ and $v_c^i(n) \geq v_c^k(n)$. Thus, after time $N_\epsilon$,

$$\frac{d(\hat{\mu}_c^i(n), v_c^i(n)) - d^+(\hat{\mu}_c^i(n), v_c^i(n) - \beta_c)}{d(\hat{\mu}_c^k(n), v_c^k(n)) - d^+(\hat{\mu}_c^k(n), v_c^k(n) - \beta_c)} \geq \frac{d(\mu_c^i - \epsilon, v_c^i(n)) - d^+(\mu_c^i - \epsilon, v_c^i(n) - \beta_c)}{d(\mu_c^k + \epsilon, v_c^i(n)) - d^+(\mu_c^k + \epsilon, v_c^i(n) - \beta_c)}$$

We also have $\mu_c^i \geq \mu_c^k$, and hence the right hand side is increasing with respect to $v_c^i(n)$.

$$\frac{d(\mu_c^i - \epsilon, v_c^i(n)) - d^+(\mu_c^i - \epsilon, v_c^i(n) - \beta_c)}{d(\mu_c^k + \epsilon, v_c^i(n)) - d^+(\mu_c^k + \epsilon, v_c^i(n) - \beta_c)} \geq \frac{\alpha_c^i}{\alpha_c^k} - O(\epsilon).$$

Thus, the inequality continues as

$$\leq N_\epsilon + \sum_{n=N_\epsilon+1}^{T} X_n.\mathbb{I}\left\{t_c^i(n)[d\left(\hat{\mu}_c^i(n), \mu^*\right) + \sum_{k \in \mathcal{K}_c \backslash \{i\}} \frac{t_c^k(n)}{t_c^i(n)}d^+\left(\hat{\mu}_c^k(n), \mu^* - \beta_c\right)] \leq \log n + a \log \log n\right\}$$

$$\leq O(1) + \sum_{n=1}^{T} X_n.\mathbb{I}\left\{t_c^i(n)[d\left(\mu_c^i + \epsilon, \mu^*\right) + \sum_{k \in \mathcal{K}_c \backslash \{i\}} \frac{\alpha_c^i}{\alpha_c^k}d^+\left(\mu_c^k + \epsilon, \mu^* - \beta_c\right)] \leq \log n + a \log \log n + O(\epsilon)\right\}$$

$$\leq O(1) + \sum_{n=1}^{T} X_n.\mathbb{I}\left\{t_c^i(n)[d\left(\mu_c^i + \epsilon, \mu^*\right) + \sum_{k \in \mathcal{K}_c \backslash \{i\}} \frac{\alpha_c^i}{\alpha_c^k}d^+\left(\mu_c^k + \epsilon, \mu^* - \beta_c\right)] \leq \log T + a \log \log T + O(\epsilon)\right\}$$

$$\leq O(1) + \sum_{n=1}^{T} \sum_{s=1}^{n} Y_n.\mathbb{I}\left\{s[d\left(\mu_c^i + \epsilon, \mu^*\right) + \sum_{k \in \mathcal{K}_c \backslash \{i\}} \frac{\alpha_c^i}{\alpha_c^k}d^+\left(\mu_c^k + \epsilon, \mu^* - \beta_c\right)] \leq \log T + a \log \log T + O(\epsilon)\right\}$$

where
$$Y_n = \mathbb{I}\left\{A_n = (i,c), t_c^i(n) = s\right\}$$

$$\leq O(1) + \sum_{n=1}^{\infty}\sum_{s=1}^{n} Y_n.\mathbb{I}\left\{sd\left(\mu_c^i + \epsilon, \mu^*\right) + \sum_{k \in \mathcal{K}_c \backslash \{i\}} \frac{s\alpha_c^i}{\alpha_c^k} d^+\left(\mu_c^k + \epsilon, \mu^* - \beta_c\right) \leq \log T + a\log\log T + O(\epsilon)\right\}$$

$$\leq O(1) + \sum_{n=1}^{\infty}\sum_{s=1}^{n} Y_n.\mathbb{I}\left\{sd\left(\mu_c^i, \mu^*\right) + \sum_{k \in \mathcal{K}_c \backslash \{i\}} \frac{s\alpha_c^i}{\alpha_c^k} d^+\left(\mu_c^k, \mu^* - \beta_c\right) \leq \log T + a\log\log T + O(\epsilon)\right\}$$

$$\leq O(1) + \sum_{n=1}^{\infty}\sum_{s=1}^{n} Y_n.\mathbb{I}\left\{s(d\left(\mu_c^i, \mu^*\right) + \sum_{k \in \mathcal{K}_c \backslash \{i\}} \frac{\alpha_c^i}{\alpha_c^k} d^+\left(\mu_c^k, \mu^* - \beta_c\right)) \leq \log T + a\log\log T + O(\epsilon)\right\}$$

$$\leq O(1) + \sum_{n=1}^{\infty}\sum_{s=1}^{n} Y_n.\mathbb{I}\left\{s\alpha_c^i L_c \leq \log T + a\log\log T + O(\epsilon)\right\}$$

$$\leq O(1) + \sum_{s=1}^{\infty}\mathbb{I}\left\{s\alpha_c^i L_c \leq \log T + a\log\log T + O(\epsilon)\right\}\sum_{n=s}^{\infty} Y_n$$

$$\leq O(1) + \frac{\log T + a\log\log T + O(\epsilon)}{\alpha_c^i L_c}$$

Since all Clus-UCBs converge to $\mu^*$ and all empirical means converge to their actual means, we have $f_i(n) = f_k(n)$ as $n$ tends to infinity. Thus, for any other arm in the cluster,

$$\limsup_{T \to \infty} \frac{\mathbb{E}[t_c^k(T)]}{\log T} = \frac{\alpha_c^i}{\alpha_c^k}\frac{1}{\alpha_c^i L_c} = \frac{1}{\alpha_c^k L_c}.$$

