# OpenReview forum: "Clus-UCB: A Near-Optimal Algorithm for Clustered Bandits"
_TMLR — Accepted by TMLR_

### Review · Reviewer_7U2N · 2025-09-07

**Summary Of Contributions:**

This paper studies a structured variant of the multi-armed bandit (MAB) problem in which arms are grouped into known clusters. The key assumption is that the mean rewards of arms within a single cluster are constrained to be within a known threshold. The authors first establish a regret lower bound that improves upon the classical result of Lai and Robbins (1985). Building on this, they propose Clus-UCB, an algorithm specifically designed to exploit the clustering structure, which asymptotically matches the new lower bound in most cases. The paper then provides empirical comparisons against KL-UCB and two versions of a Two-Level Policy (TLP). The results show that Clus-UCB consistently outperforms KL-UCB, while TLP occasionally achieves better performance on certain instances. Finally, the authors examine robustness under misspecified cluster widths and discuss the limitations of their approach.

**Audience:**

Yes

**Audience Explanation:**

The work introduces a new structured bandit formulation where arms are grouped into clusters with bounded intra-cluster differences, an assumption that has not been fully explored in prior literature. This is accompanied by both a theoretical contribution (an improved regret lower bound) and a practical algorithm (Clus-UCB) that nearly achieves this bound. Since structured bandits and clustering assumptions are an active area of research, at least part of the TMLR readership would be engaged by these findings.

**Broader Impact Concerns:**

Since this work is primarily theoretical and focuses on proposing a new algorithm for sequential decision-making, it does not raise any obvious ethical concerns.

**Claims And Evidence:**

No

**Claims Explanation:**

## Introduction

The introduction is lacking in detail. The paper introduces a structured bandit setting where the mean rewards of arms within a cluster are constrained to lie within a known threshold, and proposes an algorithm to solve this efficiently. However, if a new problem formulation is introduced, the paper should more clearly justify why this setting is needed and how it differs from existing dependent-arm formulations. Currently, there is insufficient explanation of what structured bandit problems already exist, why they are inadequate, and why a new formulation is necessary. A direct comparison with prior work, particularly Pandey et al. (2007), is essential—if their setting is a special case of the current one, then the differences should be spelled out both conceptually and mathematically, including a regret comparison.

The literature review is also too thin. For example, relevant works such as [1], [2] should be included and discussed.

[1] Singh, R., Liu, F., Sun, Y., & Shroff, N. (2024). Multi-armed bandits with dependent arms. Machine Learning, 113(1), 45-71.
[2] Gupta, S., Chaudhari, S., Joshi, G., & Yağan, O. (2021). Multi-armed bandits with correlated arms. IEEE Transactions on Information Theory, 67(10), 6711-6732.


## Analysis

The theoretical analysis is restricted to Bernoulli bandits, and the paper states that "we believe this can be extended to the exponential family." This claim needs stronger justification. In multi-armed bandit theory, analyses are typically developed for general reward distributions with finite variance, not just Bernoulli, so the current restriction makes the contribution appear less general.

The paper claims that the regret lower bound is improved, but it does not explicitly restate the original classical bound for direct comparison. A clear side-by-side comparison with the classical Lai and Robbins (1985) bound would make the contribution much clearer. Additionally, the proofs appear to rely heavily on existing frameworks (e.g., Graves & Lai (1997)), so the novelty of the analysis should be clarified. It is natural that imposing a stronger structure would reduce regret, so the precise technical innovation needs to be highlighted.

##  Experiment

The experimental evaluation is significantly underdeveloped. The experiments are conducted only on very small-scale synthetic problems (e.g., two clusters with six arms in total). This is insufficient to demonstrate robustness. Larger-scale synthetic experiments—with many clusters, varying levels of overlap, and broader parameter regimes—are needed to support the claims.

Moreover, the simulations assume that the cluster widths ($\beta_c$) are known, but in practice, this information is usually unavailable. The paper should explain how $\beta_c$ might be estimated in real-world settings. I believe a direct comparison with algorithms that do not rely on $\beta_c$ is unfair.


##  Writing and Presentation

The writing style is sometimes informal and not consistently academic. There are several minor errors in spacing, grammar, and formatting. For example, the use of a colon ":" at the end of Section 7 and Section 8 headings is unconventional.

Expositional issues are also present. For instance, when introducing the KL divergence, the paper provides only a formula without explaining its meaning or intuition in words. Similarly, in Section 2.2, the notation $\mu_c^i$ is used within formulas before it is properly defined in the text, which makes the exposition confusing.

**Requested Changes:**

Please address the issues raised above. In particular, strengthening the experimental evaluation and providing a more detailed introduction would substantially improve the paper.

---

> ### Author Response · Authors · 2025-11-11
> **Response to Reviewer 7U2N (Latest)**
>
> We sincerely thank the reviewer for their careful reading of our manuscript and for the constructive feedback. We have addressed the comments below and revised the paper accordingly.
>
> ---
>
> ### *Comment:* Larger-scale synthetic experiments—with many clusters, varying levels of overlap, and broader parameter regimes—are needed to support the claims.
>
> **Response:**
> We have added an experiment involving 20 clusters and 55 arms, covering a range of configurations including wide, narrow, overlapping, and non-overlapping clusters. The results remain consistent with our theoretical claims and are provided in the Appendix (page 12).
>
> ---
>
> ### *Comment:* The literature review is too limited. For example, relevant works such as [1] and [2] should be included and discussed.
>
> **Response:**
> We have expanded the literature review to include the mentioned works and related references. We also clarify how our setting differs from theirs in terms of assumptions and problem formulation. The revised discussion appears in Section 1.1 (page 2).
>
> ---
>
> ### *Comment:* The theoretical analysis is restricted to Bernoulli bandits, and the paper states that “we believe this can be extended to the exponential family.” This claim needs stronger justification.
>
> **Response:**
> We have extended the scope of Theorem 2 to include a broader class of reward distributions, such as bounded and canonical exponential-family distributions. The justification for this extension is now provided with references to relevant theorems and lemmas from related works. The detailed discussion can be found in Section 4 (page 6).
>
> ---
>
> ### *Comment:* The paper claims that the regret lower bound is improved, but it does not explicitly restate the original classical bound for direct comparison.
>
> **Response:**
> We now explicitly restate the classical regret bound to facilitate direct comparison with our proposed bound. This addition can be found in Section 3 (page 5).
>
> ---
>
> ### *Comment:* The paper should explain how $\beta_c$ might be estimated in real-world settings.
>
> **Response:**
> We have added a discussion on cluster width estimation in Section 2, along with a possible estimator. While estimation is not the main focus of this work, we agree that this is an important aspect for practical deployment. Our focus is on developing a lower bound and algorithm assuming known cluster widths. Nevertheless, exploring estimation under uncertain or unreliable prior information is indeed a promising direction for future work.
>
> ---
>
> ### *Comment:* Writing and presentation issues.
>
> **Response:**
> We have carefully revised the manuscript to address the mentioned issues, as well as other minor language and presentation improvements throughout the text.

---

> > ### Comment · Reviewer_7U2N · 2025-11-13
> >
> > Thank you for the revision; I appreciate your effort in addressing my comments. While some of my initial concerns are resolved, I still have questions regarding the experimental results and analysis.
> >
> > The core issue is that the explanation of the experimental results, especially the comparison with TLP, needs to be much clearer. Readers need to know when to use TLP versus Clus-UCB.
> >
> > 1. TLP Parameter Values: What specific cluster parameter values are you using for TLP in each simulation?
> >
> > 2. Figures 2, 3, 4: In the cases presented in Figures 2, 3, and 4, which I interpret as examples with a reasonable cluster structure, the performance of TLP-Mean is similar to, or isignificantly better than Clus-UCB. When the underlying structure is well-defined, why should a user choose Clus-UCB over TLP?
> >
> > 3. Figure 5: Since this figure shows Clus-UCB strongly outperforming TLP under misspecified clustering, are you arguing that Clus-UCB is significantly more robust to clustering errors than TLP?
> >
> > 4. Figure 6: The difference from Figure 5 appears to be a misspecification of $\beta$. If $\beta$ is misspecified, why does Clus-UCB's performance in Figure 6 actually improve compared to Figure 5?
> >
> > **Minor Formatting Suggestions**
> > - Please use the \ref command to point to the exact section (e.g., Appendix A.1.) both for the Appendix and for main paper sections. This greatly improves reader navigation.

---

> ### Author Response · Authors · 2025-11-14
>
> Dear Reviewer,
>
> We have added appropriate `\ref{}` references throughout the manuscript to improve readability, and we have now included the mathematical expression for the cluster reward estimates for clarity and completeness.
>
> ---
>
> ### 1. Use of Cluster Information in TLP
> The TLP algorithm relies solely on the *membership* of arms within clusters. For example, in the instance with clusters
> \([0.41, 0.42, 0.43]\) and \([0.43, 0.44, 0.45]\),
> TLP knows that arms \(1,2,3\) belong to the first cluster, and arms \(4,5,6\) belong to the second cluster.
> Importantly, TLP does **not** have access to the cluster widths.
>
> ---
>
> ### 2. Performance Comparison with Clus-UCB
> As observed, TLP outperforms Clus-UCB in the instances shown in Figures 2, 3, and 4. We believe this behavior is due to the cluster structures in these examples, which are either non-overlapping or only slightly overlapping with tight concentration.
>
> - **Non-overlapping clusters:**
>   The empirical cluster mean of the optimal cluster is usually greater than that of the suboptimal cluster, enabling TLP’s first-level decision to consistently select the optimal cluster.
>
> - **Tightly overlapping clusters:**
>   A similar effect persists, and TLP typically identifies the optimal cluster.
>
> However, when clusters are allowed to be loose(wide) or heavily overlapping, pathological cases arise. For example, with clusters
> \([0.68, 0.69, 0.67]\) and \([0.1, 0.2, 0.7]\),
> the suboptimal cluster may often have a higher empirical cluster mean, preventing TLP from adequately exploring the optimal cluster. This behavior corresponds to the pathological instance shown in **Figure 5**.
>
> ---
>
> ### 3. Correction Regarding Figures
> We apologize for the earlier confusion. Width misspecification is addressed in **Figures 4 and 6**, not Figures 4 and 5. This has been corrected in the manuscript.
>
> ---
>
> ### 4. Clarification on Figures 4 and 6
> Figures 4 and 6 correspond to the same underlying bandit instances. The only difference is in the parameter vector $\beta$:
>
> - **Figure 4:** No width misspecification.
> - **Figure 6:** The width of the suboptimal cluster is underestimated.
>
> As discussed in Section 6, such underestimation is expected to improve performance, which is reflected in the experimental results.

---

> ### Comment · Reviewer_7U2N · 2025-11-14
>
> Thank you for your latest response and for providing the additional clarity regarding the experimental results. I also appreciate the addition of appropriate \ref command, which significantly improves the readability of the manuscript.
>
> To ensure I have correctly grasped the distinction, I would like to confirm my understanding of the recommended usage for Clus-UCB versus TLP in various scenarios. Is this a correct summary of your recommendation for when to employ each method?
>
> - Clus-UCB: When clusters are allowed to be loose or heavily overlapping, and an estimate of the cluster width is available.
>
> - TLP: In other cases (e.g., when clusters are well-separated, or when the cluster width is unknown).
>
> Following this, I have one final clarification: Even if the cluster width is available, if the clusters are well-separated, is TLP still recommended than Clus-UCB?
>
> If this is indeed the case (i.e., that TLP is preferred in the well-separated scenario even when cluster width is known), it would greatly enhance the clarity and accuracy of your contribution if this specific distinction were explicitly stated in Section 5 (Experiments) of the manuscript, particularly where you illustrate the experimental results.
>
> Thank you for clarifying this key distinction.
>
>
> **Minor Typo**
>
> In the second-to-last paragraph of Section 6, the phrase "int the regret bound" should be corrected to "in the regret bound."

---

> ### Author Response · Authors · 2025-11-14
>
> Dear Reviewer,
>
> Thank you for this insightful question!
> When clusters are known to be well-separated (i.e., non-overlapping), it can be shown that the regret lower bound becomes independent of cluster widths. In such scenarios, Clus-UCB becomes overly conservative in its exploration strategy, as it is designed to handle the more challenging case where clusters may overlap. This conservatism leads to more exploration than necessary when well-separation is guaranteed.
> Therefore, TLP is indeed preferred when clusters are known to be well-separated. In contrast, Clus-UCB is specifically designed for scenarios where well-separation cannot be assumed, that is, when clusters may be loose or heavily overlapping. In these cases, the cluster width parameter becomes crucial for achieving efficient exploration.
>
> We appreciate you highlighting this point, as it clarifies the cases when one algorithm might be more suitable than the other and provides clearer guidance for practitioners.
>
> We have added subsection 5.1 to discuss this.

---

> > ### Author Response · Authors · 2025-11-14
> >
> > Specifically, in the lower bound for the non-overlapping case, the contribution from suboptimal clusters can be shown to be:
> >
> > $$
> >  \sum_{c \neq c^*} {\frac{{\mu}^{\ast}-{\mu}_c^{min}}{d(\theta_c^{min}, {\theta}^{\ast})}} $$
> >
> > Note that this bound is always as good or better(lower) than the overlapping case as one might guess. Also this bound is independent of cluster widths. Interestingly, the only contribution in the regret is from the worst arm in each cluster. This is similar to the lower bound presented in the manuscript, except no $\beta$ shift. Thus, an optimal algorithm pulls the worst arm in a cluster $O(\log T)$ times, and the other arms in the suboptimal cluster are pulled $o(\log T)$ times.
> >
> > Though we know TLP beats Clus-UCB in this scenario, we're not sure if TLP is asymptotically optimal.

---

### Review · Reviewer_n3Dy · 2025-09-19

**Summary Of Contributions:**

### Summary

This paper investigates a clustered stochastic bandit setting where, for each cluster $c$, the mean rewards of arms differ by at most a known threshold $\beta_c$. The authors first derive an asymptotic regret lower bound that improves upon the classical Lai–Robbins bound by explicitly leveraging the intra-cluster constraints. They then propose Clus-UCB, a KL-UCB-based algorithm that achieves regret matching the lower bound up to constants in most instances. Simulations against KL-UCB and two variants of the Two-Level Policy (TLP) demonstrate the empirical advantages of Clus-UCB.

### Strengths

1. The problem formulation is a natural generalization of existing models where arms in the same cluster are assumed to have identical parameters.
2. The regret lower bound explicitly captures the effect of cluster structure, especially the threshold $\beta_c$, and the proposed Clus-UCB algorithm is near-optimal.
3. The paper includes a clear and detailed theoretical discussion. For instance, Theorem 1 highlights how the proposed bound refines the classical Lai-Robbins bound, and the authors provide insightful explanations through special cases.

### Weaknesses

1. Although the problem setting is general, Theorem 2 and the main analysis are restricted to the Bernoulli case. Providing more details or extensions to other distributions would strengthen the generality claims.
2. The proofs for both Theorems 1 and 2 follow the standard frameworks from Graves & Lai (1997) and Garivier & Cappé (2011).
3. There is no textual explanation for Algorithm 1. The authors should clarify the intuition behind including KL terms from other arms with a $q - \beta_c$ shift, and explain the overall design rationale.
4. The experimental baselines are either unstructured (KL-UCB) or somewhat outdated (TLP, 2007). Including more recent structured bandit algorithms would offer a fairer and more comprehensive comparison. There are also no error bars in the plots.

**Audience:**

Yes

**Audience Explanation:**

The topic of multi-armed bandits is relevant to TMLR. The paper examines an underexplored setting that should interest researchers in stochastic bandits, recommendation, and reinforcement learning.

**Broader Impact Concerns:**

There are no broader impact concerns worth noting because this work is primarily theoretical.

**Claims And Evidence:**

Yes

**Claims Explanation:**

The theoretical claims are mostly well-supported by the lower bound (Theorem 1) and the algorithm analysis (Theorem 2). However, the generality claims (e.g., extension to the exponential family) are only mentioned briefly and not formalized. The design motivation of Algorithm 1 is also not well-justified.

While the evaluations are somewhat limited in scope (See weakness 4), they qualitatively support the theoretical claims.

**Requested Changes:**

1. Clarify Algorithm 1's design and intuition. (Critical)
2. Include more recent baselines and add error bars in the evaluation results. (Critical)
3. Extend Theorem 2 to generalized distributions beyond Bernoulli. (would strengthen the work)

---

> ### Author Response · Authors · 2025-11-11
> **Response to Reviewer n3Dy**
>
> We sincerely thank you for reviewing our work! We acknowledge the weaknesses/errors you correctly pointed out, and have made necessary changes in order to rectify and address them in the revised manuscript.
>
> ### *Comment:* Clarify Algorithm 1's design and intuition.
>
> **Response:**
> We have added a detailed intuitive explanation of our algorithm's design and working to aid understanding. This can be found in the last two paragraphs of Section 4 (page 7).
>
> ---
>
> ### *Comment:* Include more recent baselines and add error bars in the evaluation results.
>
> **Response:**
> We have rerun all simulations and now report 95% confidence intervals. While we acknowledge that the comparison involves older baselines, to the best of our knowledge, no prior work exactly matches our problem setting. The Two Level Policy (TLP) setting is the closest, hence its inclusion. The comparison with KL-UCB is also appropriate since our algorithm is inspired by it and extends it to structured cases.
>
> ---
>
> ### *Comment:* Extend Theorem 2 to generalized distributions beyond Bernoulli.
>
> **Response:**
> We have extended the algorithm to handle bounded reward distributions and the canonical exponential family. This can be found in the statement of Theorem 2 (page 6). The extension is similar to how KL-UCB extends to these families.
>
>
>
> ---
>
> ### *Comment:* The proofs for both Theorems 1 and 2 follow the standard frameworks from Graves & Lai (1997) and Garivier & Cappé (2011).
>
> **Response:**
> We agree that the analysis follows standard framework, however, we believe that the results are interesting. The lower bound achieved contains a minimization term, and the dominating factor depends on the specific instance. Further, capturing the specific nature of intra-cluster arm interaction within the algorithm is also worth highlighting.

---

### Review · Reviewer_Stnc · 2025-11-02

**Summary Of Contributions:**

The paper studies stochastic multi-armed bandits where arms are partitioned into known clusters and any two arms in a cluster differ in mean by at most a known width. It derives an asymptotic regret lower bound that is strictly tighter than the classical Lai–Robbins bound by leveraging this structure, then proposes Clus-UCB, an index policy that lets arms share statistical evidence within a cluster via a KL-based confidence term that includes other arms’ observations. The analysis shows near-optimality: Clus-UCB matches the lower bound on most instances and yields interpretable properties. Simulations against KL-UCB and a two-level cluster policy confirm lower regret and examine robustness when cluster widths are misspecified (overestimation is benign; underestimating the optimal cluster’s width can hurt). The paper concludes with limitations and extensions (e.g., exponential-family rewards, Thompson-style variants).

**Audience:**

Yes

**Audience Explanation:**

Yes—bandits are foundational in ML, and this paper’s clustered setting with tighter bounds and a near-optimal algorithm would interest TMLR readers focused on theory and decision-making systems.

**Claims And Evidence:**

Yes

**Claims Explanation:**

- They prove a tighter regret lower bound for clustered bandits than the classical Lai–Robbins unstructured case, and show Clus-UCB’s regret upper bound matches it on most instances.

- The Clus-UCB index aggregates other arms’ statistics in the same cluster via a KL term, extending KL-UCB to exploit cluster structure—so arms “share” information and explore more efficiently than structure-unaware policies.

- Experiments show Clus-UCB consistently outperforms KL-UCB and remains competitive with two-level policies; moreover, it stays strong when cluster widths are overestimated and its misspecification analysis depends mainly on the optimal cluster’s width.

**Requested Changes:**

- Motivation via real deployments.
Although the setting (overlapping clusters with width constraints) is interesting, the paper lacks real-world grounding. Please add concrete applications (e.g., taxonomy-based recommenders where items share categories/attributes; clinical dosing where treatments share pharmacological classes) and explain how cluster overlaps and β-constraints emerge operationally (how practitioners would estimate/monitor them).

- Beyond Bernoulli rewards.
Consider extensions to general exponential-family/sub-Gaussian rewards, and discuss whether the analysis/indices change. Even better, outline (or prototype) a contextual bandit variant where clusters are defined in feature space and evidence sharing uses context-conditioned similarities.

- Regret upper bounds—clarify and strengthen.
Consider providing a regret upper bound, which can place this work better in the related literature.

- Real-data evaluation.
May add experiments on some public real-world datasets, like recommender or online-decision datasets (e.g., news/recs bandit logs or clinical decision logs). Include offline policy evaluation (IPS/DR) and report sensitivity to β estimates and cluster overlap. This would substantively validate the practicality of the proposed structure.

---

> ### Author Response · Authors · 2025-11-11
> **Response to Reviewer Stnc**
>
> We sincerely thank you for your thoughtful and constructive feedback. We have revised the manuscript to address each of your comments in detail below.
>
> ---
>
> ### *Comment:* The motivation could be strengthened by including real-world deployment contexts.
>
> **Response:**
> We have added applications in E-Commerce and Clinical Trials where this framework can be effectively used. We have also introduced a possible cluster width estimator to improve interpretability and practical applicability. This can be found in Section 2.2 (page 4).
>
> However, we'd like to mention that the main focus of this work is not to provide the correct methodology of estimation, but rather to propose a framework to exploit such a structure if it is available. The idea of extending this work to contextual bandits clustered in their feature space is very interesting, and a promising future avenue to build upon this work. We will surely explore this direction.
>
> ---
>
> ### *Comment:* Beyond Bernoulli rewards. Consider extensions to general exponential-family/sub-Gaussian rewards, and discuss whether the analysis/indices change.
>
> **Response:**
> We have generalized the algorithm to distributions with bounded rewards and to members of the canonical exponential family. While the proof is presented for Bernoulli rewards for clarity, the theoretical results extend directly to these broader families. The extension and its justification can be found in Section 4 (page 6).
>
> ---
>
> ### *Comment:* Regret upper bounds—clarify and strengthen. Consider providing a regret upper bound, which can place this work better in the related literature.
>
> **Response:**
> Theorem 2 provides a regret upper bound on Clus-UCB. Could you please clarify which upper bound you mean in order to avoid any misunderstanding?
>
>
> ---
> ### *Comment:* Real-data evaluation. May add experiments on some public real-world datasets, like recommender or online-decision datasets (e.g., news/recs bandit logs or clinical decision logs).
>
> **Response:**
> We appreciate the reviewer’s valuable suggestion. We fully agree that incorporating real-world datasets would further strengthen the empirical validation of our approach. However, we believe that the current set of experiments, designed to closely emulate practical clustered settings, already substantiates the key claims and theoretical insights presented in this work. We plan to explore real-world evaluations as an important direction for future work.

---

### Comment · Action_Editor_Fno4 · 2025-11-03
**Author discussion started**

Dear authors,

Thank you for submitting the manuscript and for responding to Reviewer 7U2N's initial review. Please read carefully the other two reviewers' comments and discuss whether there are any potential misunderstandings. The goal of this period is to avoid possible confusion and to reach the clearest decision. After two weeks, the reviewers can submit the recommendation.

> Reviewer 7U2N,

Please let us know if you have further comments on the reply and the revision by the authors.

Best,
AE

---

### Comment · Action_Editor_Fno4 · 2025-12-01
**Errors in the proof in Theorem 2**

Dear authors,

Sorry for running behind schedule. I think all reviewers are okay with the contribution. Since there are existing concerns on the completeness of the results (Reviewer Stnc "Regret upper bounds—clarify and strengthen. Consider providing a regret upper bound, which can place this work better in the related literature.", Reviewer 7U2N "Analysis" section), I also took a look at the main results.

* Theorem 1: The lower bound of the regret with its proof in Appendix A.2 makes sense.
  * Essentially, the bandit optimization is linear semi-infinite programming (linear program with infinite number of constraints). After identifying a finite number of active constraints, it is reduced to a linear program.
* Theorem 2: The upper bound of the proposed algorithm with its proof in Appendix A.3., in my view, is incomplete.

Given these observations, I would like the authors to revise the paper to complete the proof before I make the final decision.

In the following, I suggest why the proof in Theorem 2 is incomplete.
* p14, Theorem 3 statement : For all $\delta > k+1$ seems to make the theorem useless, as we typically want to choose small $\delta$ whereas this requires $\delta$ to be larger than $1$.
* p14, Theorem 3 is not very useful as long as it is for each fixed $n$. I guess it is $P[\bigcap_n …]$.
* The first display equation there is no index  $i$ in the summed values.
 * "In the following proof, we haven’t explicitly mentioned the dependence of empirical means and number of pulls on the time step n. This is done only for neatness." I think this is too informal to put in a proof.
* "$O(\log\log T)$ as proved ahead"; this is very nontrivial. Please derive the bound explicitly. The next "Note that" shows the quantity that does not directly match $v^*(n)$.
* p15–16: I guess many outer square brackets like $[1${$...$}$]$ can just be removed.
* p15 last displayed equation $t_c^i$ in the denominator is misplaced?...
* p16 "1. Regret upper bound 2. Convergence of Clus-UCB values": These are not appropriately placed - these do not help complete the proof.
* p16. "Now, notice that".. the transformation is wrong? $A = B$ does not imply $A-C=B$. The index in the summation ($\sum$) in the second equation is missing.
* p16 "since we are interested in the time instances…$g_i(n) \ge g_k(n)$ is wrong? If $f_k(n)+g_k(n)$ is the UCB index, then $f_i(n)+g_i(n) \ge $f_k(n)+g_k(n)$ though.
* p.16 "By the strong law of… only finitely many times": It is true if you count by each sample size, however, it is not trivial if you count the number of rounds. For example, if you run a greedy algorithm, then the expected number of rounds where the best arm is underestimated is $\infty$.
* p17 $\lim$ cannot be used unless you bound from below (use $\limsup$ if you only bound from above).
* I could not check p17, but given there are already many suggestions above, let me check after you revise the paper once.

Hope this helps in improving the paper. I appreciate the opportunity to participate in the review of the paper.

Best,

AE

---

> ### Author Response · Authors · 2025-12-14
>
> Dear AE,
>
> Thank you for your careful review of our manuscript and for the insightful comments. Your feedback has helped us improve both the clarity and rigor of the presentation. We have revised the manuscript accordingly, and we address your points below. Further, all the formatting related issues pointed out have been corrected.
>
> ---
>
> ### 1) p.14, Theorem 3 statement
>
> **Comment:** *For all $\delta > k+1$ seems to make the theorem useless, as we typically want to choose small $\delta$.*
>
> **Response:**
> Here, $\delta$ is a representational parameter and is not intended to be chosen as a small constant. In fact, choosing $\delta$ to be small would lead to a trivial or vacuous concentration bound. In our analysis, $\delta$ effectively represents a $\log n + \log \log n$ term. This choice aligns directly with the form of the confidence bounds used in our UCB index. Consequently, the theorem is meaningful and useful in the context in which it is applied.
>
> ---
>
> ### 2) p.14, usefulness of Theorem 3 for fixed $n$
>
> **Comment:** *Theorem 3 is not very useful as long as it is for each fixed $n$.*
>
> **Response:**
> The theorem is indeed useful, as it is applied while summing over all rounds indexed by $n$. This summation arises naturally in the regret analysis of the algorithm. The way in which Theorem 3 is used in this cumulative manner is made explicit in the analysis presented on p.15 of the revised manuscript.
>
> ---
>
> ### 3) p.14, claim of $O(\log \log T)$
>
> **Comment:** *“$O(\log \log T)$ as proved ahead” is very nontrivial. Please derive the bound explicitly.*
>
> **Response:**
> We now provide a clearer derivation of this bound. The $O(\log \log T)$ term follows directly from Theorem 3, combined with the structural information of the problem. Specifically, the fact that arm means within a cluster cannot differ by more than $\beta_c$. We have revised the proof to make this dependency explicit and to improve readability.
>
> ---
>
> ### 4) p.15, last displayed equation
>
> **Comment:** *The term $t_c^i$ in the denominator appears to be misplaced.*
>
> **Response:**
> The term is correctly placed. We factor $t_c^i$ out and divide by it inside the bracketed expression. We have rechecked the derivation and clarified the presentation to avoid confusion.
>
> ---
>
> ### 5) p.16, placement of results
>
> **Comment:** *“1. Regret upper bound 2. Convergence of Clus-UCB values” are not appropriately placed and do not help complete the proof.*
>
> **Response:**
> We believe these results are important, as they characterize the asymptotic behavior of the proposed algorithm. In particular, they establish that the algorithm is uniformly good and highlight its improvement over KL-UCB. For these reasons, we have retained these results and clarified their role in the overall analysis.
>
> ---
>
> ### 6) p.16, transformation step
>
> **Comment:** *“Now, notice that …” the transformation appears to be wrong.*
>
> **Response:**
> We understand the source of the confusion. The missing index was intended to indicate summation over all $k$, without additional constraints such as $k \neq i$. We have now explicitly added the summation index in the revised version to remove any ambiguity.
>
> ---
>
> ### 7) p.16, definition of the UCB index
>
> **Comment:** *$f_k + g_k$ is not the UCB index.*
>
> **Response:**
> This is correct. The term $f_k + g_k$ is an intermediate quantity used in the calculation of the index. The actual UCB index is $v_c^i$.
>
> ---
>
> ### 8) p.16, use of the Strong Law of Large Numbers
>
> **Comment:** *The claim “only finitely many times” is not trivial when counting the number of rounds.*
>
> **Response:**
> We show that all arms are explored infinitely often and that no arm is starved. This follows from the KL-UCB–type upper bound, which ensures an $O(\log T)$ number of pulls for suboptimal arms. As a result, the sample size of each arm grows unbounded, allowing us to invoke the Strong Law of Large Numbers. In the counterexample you mention (e.g., a greedy algorithm), an arm may stop being pulled with positive probability, causing its sample size to remain finite. This situation does not arise in our setting.
>
> ---
>
> We sincerely thank you again for your constructive feedback, which has significantly improved the quality of the manuscript.

---

> > ### Comment · Action_Editor_Fno4 · 2025-12-17
> >
> > Dear authors,
> >
> > Thank you for your reply and revision. It seems that some parts of the questions are from my misunderstanding, and clarification helps. However, I still have several questions:
> >
> > > "since we know that all arms are pulled infinitely often asymptotically $(O(\log T))$"
> >
> > Could you point out the point where this is formally derived? It implies a lower bound on the draw, but usually such a result is non-trivial. Moreover,
> >
> > > "Since we know that all arms are pulled infinitely often asymptotically, by the strong law of large numbers, the empirical mean of an arm differs from its true mean by more than $\epsilon$ only finitely many times."
> >
> > I am not sure about the correctness because there might be events that occur infinitely often almost surely, but the expected time until the first event is infinite (and thus the empirical mean of an arm differs from its true mean for infinitely many rounds in expectation).
> >
> > Best,
> >
> > AE

---

### Decision · Action_Editor_Fno4 · 2026-01-12

**Recommendation:** Accept with minor revision

**Additional Comments:**

Please address the following minor issues. I think these are relatively easy to address.

* p3 admissible: Admissibility in statistical decision theory refers to a policy that is not dominated by any other policy. Maybe it is "adaptive" to the history?
* p4 $C(\theta)$ the second summation $\sum_{k \ne k^*}$ only runs for $k$ in the optimal cluster. And the first summation from $k=1$ to $K_c$ assumes the arms in each cluster are indexed from one to $K_c$. I think in this case the paper should clarify the range and use the notation appropriately. This applies to many other places.
* p6 Algorithm 1 line 10: Pull arm $k$ in cluster $c$.
* Please run the experiment again with $\log\log n$ forced exploration, since the algorithm has changed from the previous revision. Alternatively, clarify that it omits lines 9-10 og Algorithm 1 in the experiment.

**Audience:**

Yes

**Audience Explanation:**

The reviewers are overall interested in the paper. Online learning is one of the most popular topics and I am sure this paper attracts significant fraction of interests.

**Claims And Evidence:**

Yes

**Claims Explanation:**

This paper studies a version of the multi-armed bandit problem in which the arms are clustered (known), and two arms in the same cluster have similar means. The author proposed a version of KL-UCB and showed its performance exceeded the original (unstructured) KL-UCB. Reviewers generally find this paper is not revolutionary but fine and interesting for its analysis of considering efficient exploration that is beyond Lai and Robbins' classical bound.

I have had several discussions with the authors regarding the proof of Theorem 2 (upper bound). In particular, they introduce a $O(\log\log t)$ forced exploration and time $N_\epsilon$ (p17, revision at Jan 12th, 2026), which indicates the time until all empirical means are $\epsilon$-close. It seems that $N_\epsilon$ is independent of $T$, and my concern was addressed.

---

> ### Comment · Action_Editor_Fno4 · 2026-02-06
> **Request for revision**
>
> Dear authors,
>
> Please submit the revised manuscript so that we can proceed with the publication process.
>
> Best,
>
> AE